# Elliott State Research Forest Timber Cruise, Oregon, 2015–2016

Todd West [ID] and Bogdan M. Strimbu *[ID]

Department of Forest Engineering and Resource Management, Oregon State University, Corvallis, OR 97331, USA; todd.west@oregonstate.edu
* Correspondence: bogdan.strimbu@oregonstate.edu

**Abstract:** The Elliott State Research Forest comprises 33,700 ha of temperate, Douglas-fir rainforest along North America's Pacific Coast (Oregon, United States). In 2015, naturally regenerated stands at least 92 years old covered 49% of the research area and sawtimber plantations younger than 68 years another 50%. During the winter of 2015–2016, a forest wide inventory sampled both naturally regenerated and plantation stands, recording 97,424 trees on 17,866 plots in 738 stands. The resulting dataset is atypical for the area as plot locations were not restricted to upland, commercially harvestable timber. Multiage stands and riparian areas were therefore documented along with plantations 2–61 years old and trees retained through clearcut harvests. This dataset constitutes the only open access, stand-based forest inventory currently available for a large area within the Oregon Coast Range. The dataset enables development of suites of models as well as many comparisons across stand ages and types, both at stand level and at the level of individual trees.

**Keywords:** Elliott State Forest; timber cruise; tree height; tree diameter; Douglas-fir; red alder; western hemlock; bigleaf maple; Oregon myrtle; western redcedar





## 1. Summary

Forest inventories measure trees with the objective of quantifying various forest attributes. A common type of forest inventory is timber cruising, which samples trees' species, height, diameter, and other characteristics with the primary intent of estimating the amount of merchantable wood present in one or more stands of trees. In the case of the Elliott State Forest, a timber cruise was needed to estimate the entire forest's financial value and assess its potential for conversion from timber production to experimental management as the Elliott State Research Forest. This uncommon measurement need led to the collection, from October 2015 to February 2016 CE, of a cruise dataset providing a snapshot of 17,700 ha of the Elliott State Forest. Of those 17,700 ha, this data descriptor describes measurements on the 16,000 ha that became part of the Elliott State Research Forest in 2022 CE.

The Elliott State Research Forest consists of a rugged complex of ridges and valleys 10–32 km inland from the Pacific Ocean in Oregon's Coast Range (Figure 1). The Elliott Forest is located along North America's west coast in the central Pacific Temperate Rainforest, containing both intensively managed plantations and reserve areas with less intense anthropogenic disturbances [1]. On plantations, merchantable tree species have been regenerated through planting since circa 1950 CE, increasingly with seedlots genetically selected for sawtimber production. Plantation trees are usually harvested before exceeding 45 m in height and 75 cm DBH (diameter at breast height, 1.37 m above ground) with a limited number of trees being retained through clearcut harvest rotations [2]. Reserve areas typically conserve 130–175 year old cohorts, which regenerated naturally following patchy

wildfires, from nearby parent trees potentially over 400 years old [3]. The resulting multiage stands contain trees ranging from shade-tolerant seedlings to shade-intolerant, dominant individuals 60–80 m tall and 100–200 cm in DBH. Wildtype individuals of the three primary plantation species (coast Douglas-fir (*Pseudotsuga menziesii* var. *menziesii* (Mirb.) Franco), red alder (*Alnus rubra* Bong.), and western hemlock (*Tsuga heterophylla* (Raf.) Sarg.) are present in the Elliott's reserves, providing genetic contrast to plantation stock. The three most abundant nonplantation species (bigleaf maple (*Acer macrophyllum* Pursh), Oregon myrtle (*Umbellularia californica* Hook. & Arn.), and western redcedar (*Thuja plicata* Donn ex D. Don) occur throughout the forest. Trees on the Elliott Forest therefore exhibit a broad range of growth responses to different stand structures. These responses are recorded implicitly through timber cruising's measurements of height–diameter relationships, stem taper, amount of live foliage, injuries sustained, stand composition, and stand density. The timber cruise's individual tree measurements also capture limits to height growth that may not be accessible in datasets limited to trees younger than 100 years old.

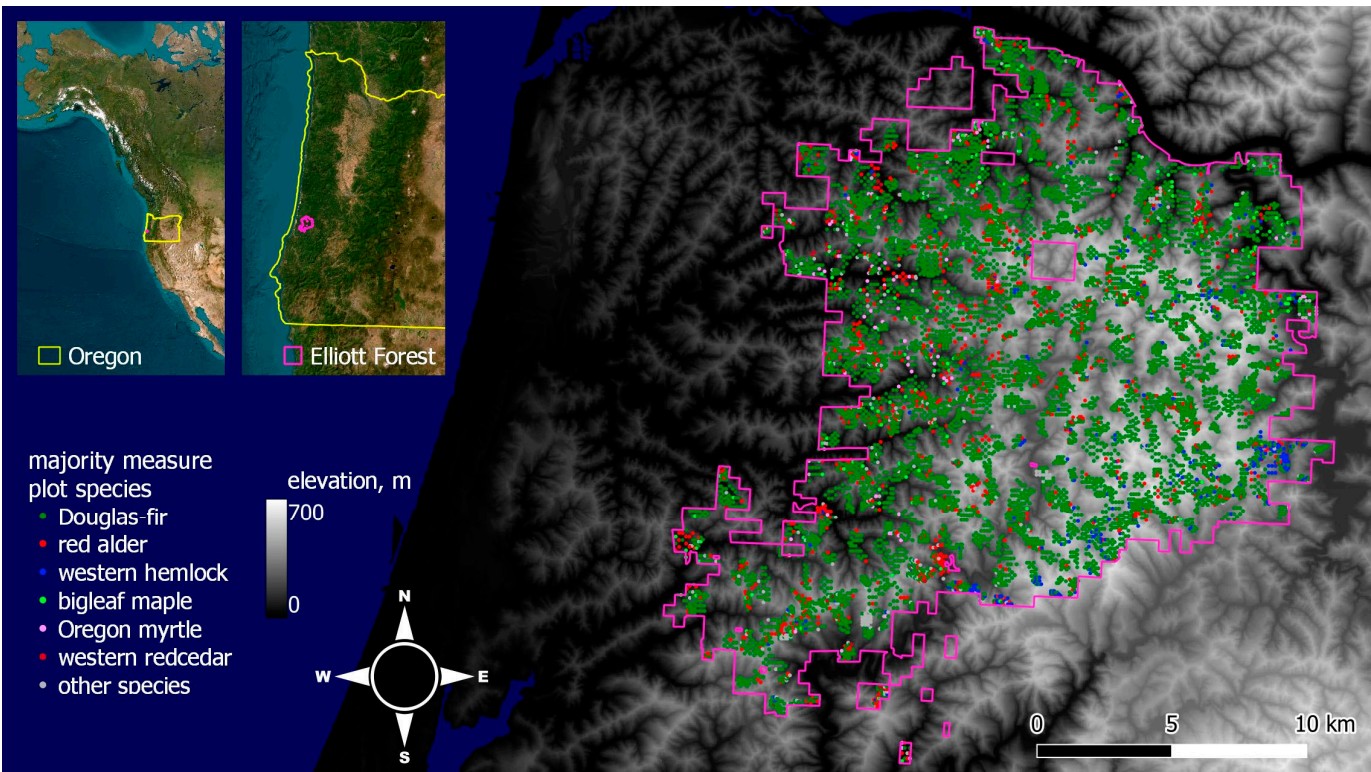

**Figure 1.** Location and topography of the Elliott State Research Forest and locations of the winter 2015–2016 timber cruise's 10,036 measure plots. The 7830 count plots were placed between measure plots in stands larger than 8.9 ha and at least 20 years old. Count plots are omitted from the figure for clarity. The primary tree species in the forest are coast Douglas-fir (*Pseudotsuga menziesii* var. *menziesii* (Mirb.) Franco), red alder (*Alnus rubra* Bong.), western hemlock (*Tsuga heterophylla* (Raf.) Sarg.), bigleaf maple (*Acer macrophyllum* Pursh), Oregon myrtle (*Umbellularia californica* Hook. & Arn.), and western redcedar (*Thuja plicata* Donn ex D. Don). See Table 1 (Section 2) for other species.

Other than this dataset, the primary open source for tree measurements in the Oregon Coast Range is the United States Forest Service's Forest Inventory and Analysis (FIA) Program [4]. As a nationwide forest inventory, the FIA Program measures trees within one 0.4 ha cluster of plots per 2428 ha of forested area in the conterminous United States [5]. FIA sampling is therefore spatially much sparser than the timber cruise data presented here, which use one approximately 0.03 ha plot per 0.4–1.6 ha of stand area. The sparseness of FIA data prevents calculation of common stand-level properties. Stand-level properties, such as the number of trees per hectare or volume of merchantable wood (m$^3$ ha$^{-1}$), are

foundational to many forest inventory methods and rely on the definition of a stand as a reasonably homogenous group of trees to be meaningful [6]. Since stands on the Elliott Forest range from about 0.4 ha up to 141 ha (median size 14 ha, mean size 18 ha) sampling intensities on the order of one plot per hectare are needed to adequately characterize stands and provide sufficient statistical power to meet timber cruise accuracy targets [2]. The Elliott's range of stand sizes is typical of the Oregon Coast Range and also of silviculturally managed areas across much of the area covered by the Northwest Forest Plan in northern California, western Oregon, and western Washington. Elliott cruise data is therefore of interest as a regional reference for design of large area timber cruises. Elliott data also captures past management influences on stand structures, biodiversity, and tree size in comparison to older stands with limited to no history of timber harvest (see [7] for a summary of the Northwest Forest Plan and regional management concerns).

Timber cruise data from the Elliott Forest thus enables a range of research questions to be posed and answered. One such research category is forest biometrics analyses of trees' allometry, focusing on relationships among height, diameter, taper, stand structure, and physiography. Another set of research topics centers on silvicultural choices in forest management, both retrospectively and in identifying how future management might select among tradeoffs in desired stand structures, biodiversity, carbon sequestration, and wood fiber. Other possibilities include topics such as integration of large area ground control datasets with remote sensing and forest dynamics as a function of tree species prevalence. Open cruise data is also valuable as an educational resource, providing forestry students a representative sample of stands to study and enabling owners of small forestlands to place their lands into a broader context.

## 2. Data Description

The Elliott State Research Forest is a Douglas-fir rainforest and, consequently, Douglas-fir was the most abundant tree species in the 2015–2016 timber cruise (Table 1). Douglas-fir dominance in the Oregon Coast Range occurs naturally as it is a comparatively rapidly growing, shade-intolerant species that occupies the overstory after stand replacing wildfire or blowdown. Douglas-fir dominance also results from management intent. The species yields high value sawtimber and is therefore preferentially planted after clearcut harvests. Some Elliott stands also contain significant fractions of red alder, western hemlock, bigleaf maple, Oregon myrtle, and western redcedar (Figure 2). In 2016, stand ages across the Elliot Forest ranged from clearcut and not yet replanted up to an estimated 226 years. The cruise sampled stands from 3–196 years old (mean plantation age 35 years, mean naturally regenerated age 131 years).

The 738 stands cruised on the Elliott State Research Forest totaled 15,986 hectares, 47% of the forest's area. Both height and diameter at breast height (DBH, 1.37 m) were measured for 30,671 live trees with intact stems, along with diameter and height to stem break for another 943 trees with broken tops. Selected trees were ocularly assessed for crown ratio to the nearest 10%, measured for taper, or cored for age at breast height (Table 1). Snags (standing dead trees) were also recorded and measured for DBH. Since cruising used English units, heights were recorded to the nearest foot (30 cm) and DBH outside bark to the nearest 0.1 inch (2.5 mm). Depending on the cruiser, diameters smaller than 4–8 inches (10–20 cm) were sometimes recorded to the nearest inch (2.5 cm). Among cruised trees, 46% were in plantations established after 1946 CE whereas 54% of cruised area was in plantation stands.

The smallest DBH class recorded was 2.5 cm and the shortest trees recorded were 1.5 m tall. Maximum sizes varied with species and sampling coverage but often exceeded 35–60 m in height and 100 cm DBH. Asymptoticity in height growth was apparent in tree species with sufficient measurements, arguably with the exception of western hemlock. Observed DBHs up to 241 cm did not show evidence of a maximum limit to diameter growth (Appendix A, Figures A1–A4).

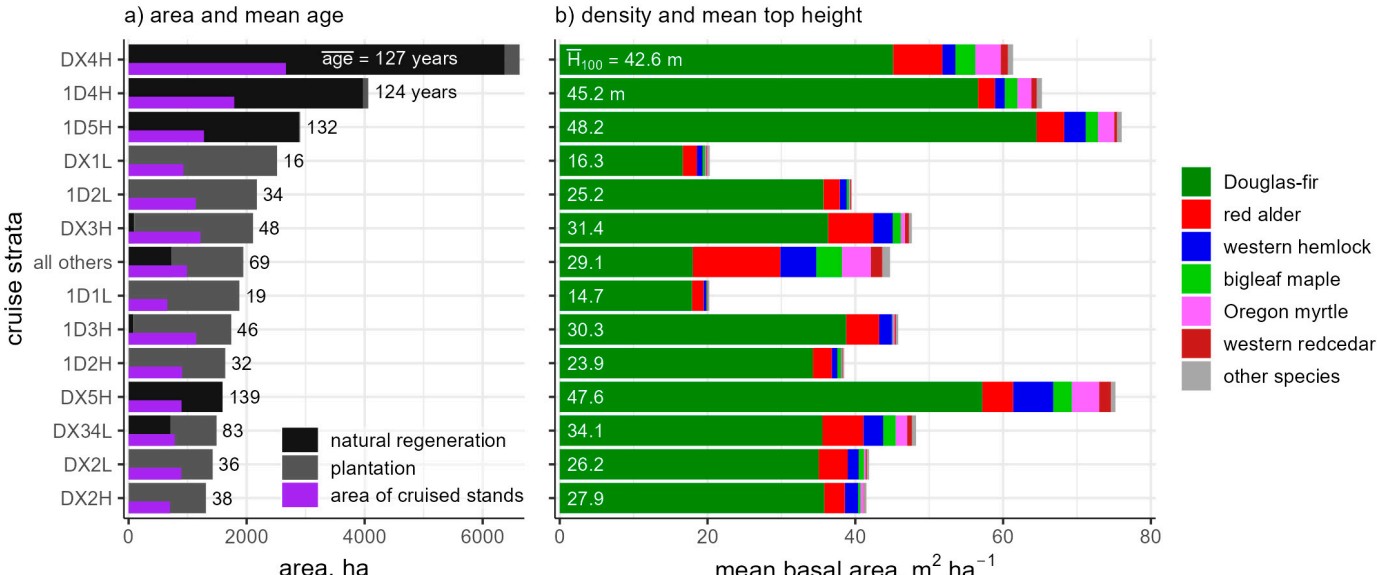

**Figure 2.** Summary of the stratification used for randomly selecting the 738 stands cruised on the Elliott State Research Forest during the winter of 2015–2016. Strata names consist of a prefix (1D or DX, DX indicating slightly less Douglas-fir dominance), pre-cruise estimate of tree size (1–5, larger numbers indicating larger size classes), and a pre-cruise estimate of density (L or H for low or high, respectively). Stand ages are as of 2016 and $\overline{H_{100}}$ is the area weighted, mean top height ($H_{100}$, m, mean height of tallest 100 trees per hectare) of the stands in each cluster. Plantations are stands whose estimated origin year is 1947 CE or later.

**Table 1.** Ground measurements collected during winter 2015–2016 timber cruising. All trees that were measured for height, whether with intact stems or broken tops, were also measured for DBH (diameter at breast height, 1.37 m) and all snags were measured for DBH. Taper measurements consist of a second diameter at a recorded height, usually 30% of tree's height. Dead trees include both recently deceased trees and decayed, but still standing, snags.

| Common Name | Live Trees Counted | Fraction of Stems, % | DBH Measured | Unbroken Height | Height to Broken Top | Crown Ratio | Taper | Breast Height Age | Dead Trees |
|---|---|---|---|---|---|---|---|---|---|
| Douglas-fir | 73,022 | 77.0 | 39,289 | 20,548 | 646 | 19,316 | 16,279 | 2079 | 1961 |
| red alder | 9163 | 9.6 | 5042 | 3780 | 192 | 3012 | 2130 | 22 | 185 |
| western hemlock | 4813 | 5.0 | 2787 | 2152 | 19 | 1575 | 1331 | 38 | 40 |
| bigleaf maple | 2654 | 2.7 | 1345 | 1047 | 34 | 579 | 383 | 0 | 18 |
| Oregon myrtle | 2600 | 2.7 | 1419 | 1062 | 26 | 806 | 533 | 0 | 18 |
| western redcedar | 1065 | 1.1 | 597 | 519 | 8 | 404 | 316 | 0 | 23 |
| other | 1719 | 1.9 | 1332 | 1266 | 8 | 261 | 189 | 0 | 143 |

The most common species on the Elliott State Research Forest are Douglas-fir (*Pseudotsuga menziesii* var. *menziesii* (Mirb.) Franco), red alder (*Alnus rubra* Bong.), western hemlock (*Tsuga heterophylla* (Raf.) Sarg.), bigleaf maple (*Acer macrophyllum* Pursh), Oregon myrtle (*Umbellularia californica* Hook. & Arn.), and western redcedar (*Thuja plicata* Donn ex D. Don). Other tree species are trees recorded as other hardwoods (0.477% of stems, species not specified), cascara buckthorn (*Rhamnus purshiana* DC., 0.468%), Pacific madrone (*Arbutus menziesii* Pursh, 0.227%), Sitka spruce (*Picea sitchensis* (Bong.) Carrière, 0.192%), cherry (*Prunus* spp, 0.142%), simply as other (0.093%), other conifers (0.092%), giant chinkapin (*Chrysolepis chrysophylla* var. *chrysophylla* (Douglas ex Hook.) Hjelmq., 0.055%), Pacific dogwood (*Cornus nuttallii* Audubon, 0.044%), tanoak (*Notholithocarpus densiflorus* (Hook & Arn.) Manos, C.H. Cannon & S. Oh, 0.043%), Port-Orford-cedar (*Chamaecyparis lawsoniana* (A. Murray) Parl., 0.038%), Pacific yew (*Taxus brevifolia* Nutt., 0.010%), grand fir (*Abies grandis* (Douglas ex D. Don) Lindl., 0.009%), black cottonwood (*Populus balsamifera* L., 0.005%), Oregon ash (*Fraxinus latifolia* Benth., 0.004%), four willows (*Salix* spp, 0.004%), two Oregon white oaks (*Quercus garryana* Douglas ex Hook, 0.002%), and a single coast lodgepole pine (*Pinus contorta* var *contorta* Douglas ex Loudon, 0.001%).

## 3. Methods

As timber cruise procedures and subsequent calculations are well documented and follow similar sampling approaches across many forest inventories (e.g., [6,8–11]), we

briefly summarize the specific cruising process used in the 2015–2016 timber cruise [2] on the Elliott Forest.

1. Using pre-existing forest inventory data on stand ages, tree species composition, and density, the 1903 stands delineated across the Elliott State Forest were stratified and a subset of stands in each strata was randomly selected for cruising (Figure 2).

2. Square grids of target plot coordinates were placed over selected stands. Stands older than 20 years were assigned a combination of count and measure plot densities of 0.62–2.47 plots ha$^{-1}$, progressively increasing in stands smaller than 44.5 ha (Figure 3). Plot densities were the same in stands younger than 20 years, but only measure plots were used.

3. Cruisers went to each stand, selected a BAF (basal area factor, also called prism factor) for the stand, traveled to each plot's target coordinates, and collected the measurements for that plot type. BAFs were chosen with the objective of having typically 5–8 trees per plot, resulting in 83% of plots having 3–10 trees.

4. In parallel with the primary cruising effort, data went through two layers of office review for correctness and 5% of plots were check cruised. Corrections were made to resolve errors found, returning to the field if necessary.

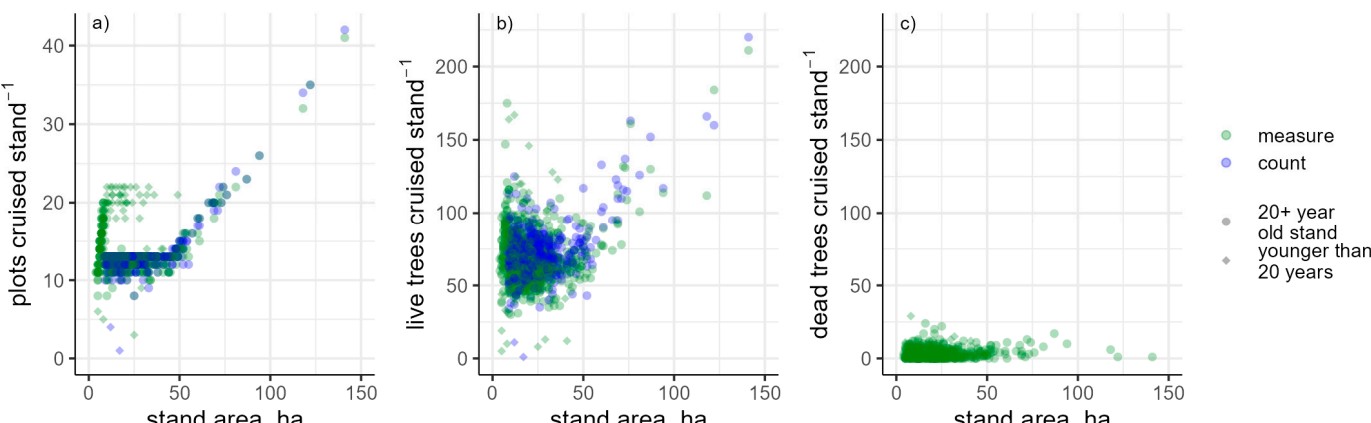

**Figure 3.** Sampling intensity in terms of (**a**) plots and (**b**,**c**) live and dead trees measured or counted in each stand cruised. Only live trees were counted on count plots. Fifty-three percent of all stems counted were measured for DBH and 32% were measured for height.

All count plots were variable radius plots where only a count of the trees in the plot, by species, was recorded. Measure plots used a nested plot design consisting of a variable radius plot for trees larger than 11.7 cm DBH and a fixed 2.54 m radius plot for trees 2.5–11.7 cm DBH. All trees on measure plots were measured for DBH and, nominally, the first and third stem of each species encountered as the cruiser swung the plot were measured for height. Height measure tree selection could be adjusted on a plot to cover a broader diameter range, if needed. Height measure trees were also measured for taper, had their compacted live crown ratios estimated, and were assessed for overall condition and form. The combination of count and measure plots forms a double sampling approach [6], allowing more accurate tree count estimates while constraining measurement effort.

A requirement on making the 2015–2016 timber cruise data openly available was plots' target coordinates and stand locations be excluded due to conservation considerations and sociopolitical factors. This restriction precludes fully spatial analyses of the cruise, but physiographic information remains available. Since plots were not stem mapped during cruising, plot target coordinates were used to calculate elevation, slope, and aspect at the plot level from a DTM (digital terrain model [12]) with 0.91 m resolution using zonal means (QGIS 3.22.9) over a nominal 10 m plot radius. Since each pixel of a DTM has a distinct aspect, independent means were taken over sin(aspect) and cos(aspect) and combined to

estimate an overall aspect for each plot. A fourth physiographic predictor, topographic shelter index, was also calculated as the absolute value of mean horizon angle in degrees to a 100 m radius over the eight azimuths 0°, 45°, ..., 315° using r.horizon (GRASS GIS 7.8.6). This calculation operates by r.horizon casting a ray from plot center along each azimuth to the specified distance and using nearby terrain, as described by the DTM, to determine the angle at which the ray intersects the local horizon. Averaging the eight azimuths produces a statistic with a value near zero in exposed locations, such as summits or ridgetops, and which increases at midslope and bottom of valley positions. Across much of the Elliott, the 100 m radius for topographic shelter index corresponds to roughly 2–3 tree heights. The topographic shelter index thus provides an approximate proxy variable for slope position.

## 4. User Notes

The Elliott 2015–2016 timber cruise dataset is provided at the individual tree measurement level. Each of the 97,424 trees measured has unique plot and stand IDs indicating which of the 17,866 plots and 738 stands the tree is located in, along with the plot's type and expansion factor. A separate file provides stand ages and areas. Additional inputs for models that use plot or stand-level variables and also comparisons across stand ages and types can thus be calculated from the cruise data using standard forest inventory methods.

**Author Contributions:** Conceptualization, T.W. and B.M.S.; methodology, T.W.; software, T.W.; validation, T.W.; formal analysis, T.W.; investigation, T.W.; resources, B.M.S.; data curation, T.W.; writing—original draft preparation, T.W.; writing—review and editing, T.W. and B.M.S.; visualization, T.W.; supervision, B.M.S.; project administration, B.M.S.; funding acquisition, B.M.S. All authors have read and agreed to the published version of the manuscript.

**Funding:** This research was funded by the Oregon Department of State Lands and Elliott State Research Forest (no grant numbers issued) and U.S. Department of Agriculture grant 2019-67019-29462.

**Institutional Review Board Statement:** Not applicable.

**Informed Consent Statement:** Not applicable.

**Data Availability Statement:** This study's dataset can be accessed at https://doi.org/10.7267/xp68kq79g (accessed on 15 January 2024).

**Acknowledgments:** The Elliott State Forest 2015–2016 timber cruise and cruise report [2] were funded by Oregon Department of State Lands RPF 141-1161-15. Further analysis, including dataset publication, was funded by Oregon State University on behalf of the Elliott State Research Forest. Edie Knight, the lead forester on the timber cruise, reviewed this data descriptor's initial draft, and provided the target cruise plot coordinates needed to include physiographic information in the dataset, but, due to time constraints, was ultimately unable to join as an author.

**Conflicts of Interest:** The authors declare no conflicts of interest. The funders had no role in the collection, analyses, or interpretation of data or in the writing of the manuscript; or in the decision to publish the results. The timber cruise was designed to meet the Oregon Department of State Lands' inventory needs (stand-level sawtimber accuracy of ±20 at 80% confidence and ±3 at 95% confidence over 33,400 ha of the project area).

## Appendix A

Figures A1–A4 show height and diameter distributions for the Elliott State Research Forest's six most abundant tree species (Figures A1–A3) and, collectively, the other 18 species, genera, and minimally identified stems in the 2015–2016 timber cruise records (Figure A4). Transitions toward asymptotic height growth occur as height–diameter ratios decline in larger Douglas-fir, red alder, western hemlock, and bigleaf maple. However, this trend reverses in western hemlock larger than 85 cm DBH (Figure A2a). The reversal may be an artifact of limited sampling of large hemlocks (n = 167) or may result from differing histories of stand establishment, disturbance, and succession not captured in this dataset.

For a given DBH, the distribution of individual trees' heights is broadly symmetrical about the mean tree height. For a given tree height, however, the distribution about mean DBH is more likely to be asymmetrical, particularly in the three most abundant broadleaf species (red alder, bigleaf maple, and Oregon myrtle). The asymmetry toward stems with comparatively large DBHs follows species' growth habits and may also reflect increased survival among undamaged individuals with lower height–diameter ratios.

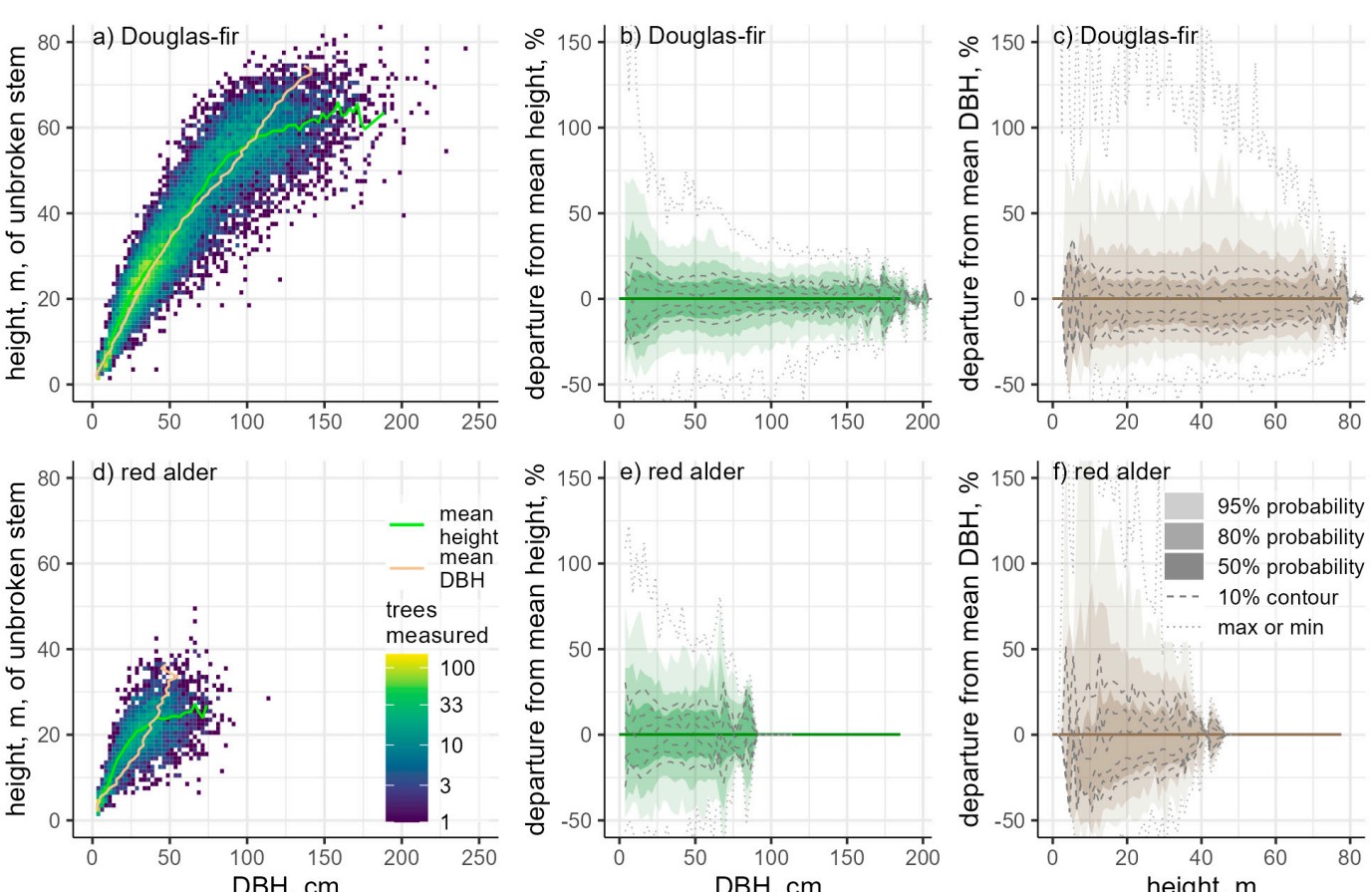

**Figure A1.** Distribution of unbroken (**a**) Douglas-fir and (**d**) red alder stem sizes in 2015–2016 Elliott State Research Forest cruise data with respect to the trees' mean height (**b**,**e**) and diameter (**c**,**f**). Lines for mean height and mean diameter stop when there are fewer than 10 trees per 1 m height class or 2.5 cm diameter class.

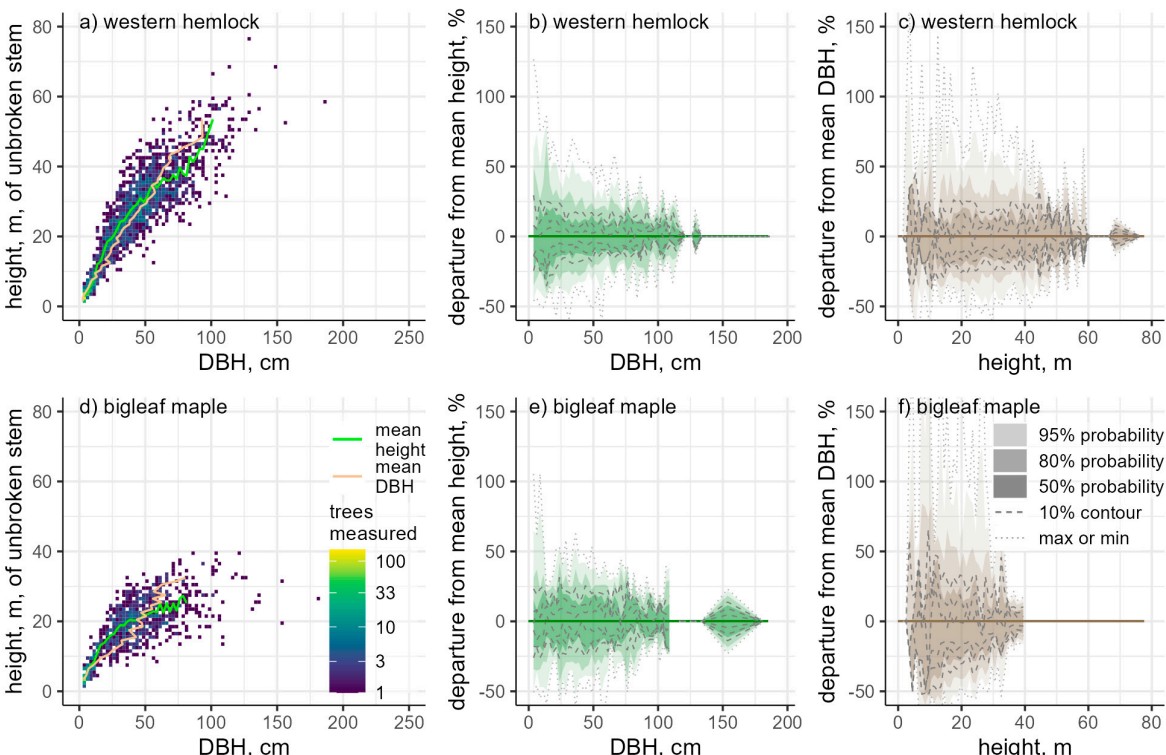

**Figure A2.** Distribution of unbroken (**a**) western hemlock and (**d**) bigleaf maple stem sizes in 2015–2016 Elliott State Research Forest cruise data with respect to the trees' mean height (**b**,**e**) and diameter (**c**,**f**). Mean height and diameter lines are drawn as indicated in Figure A1.

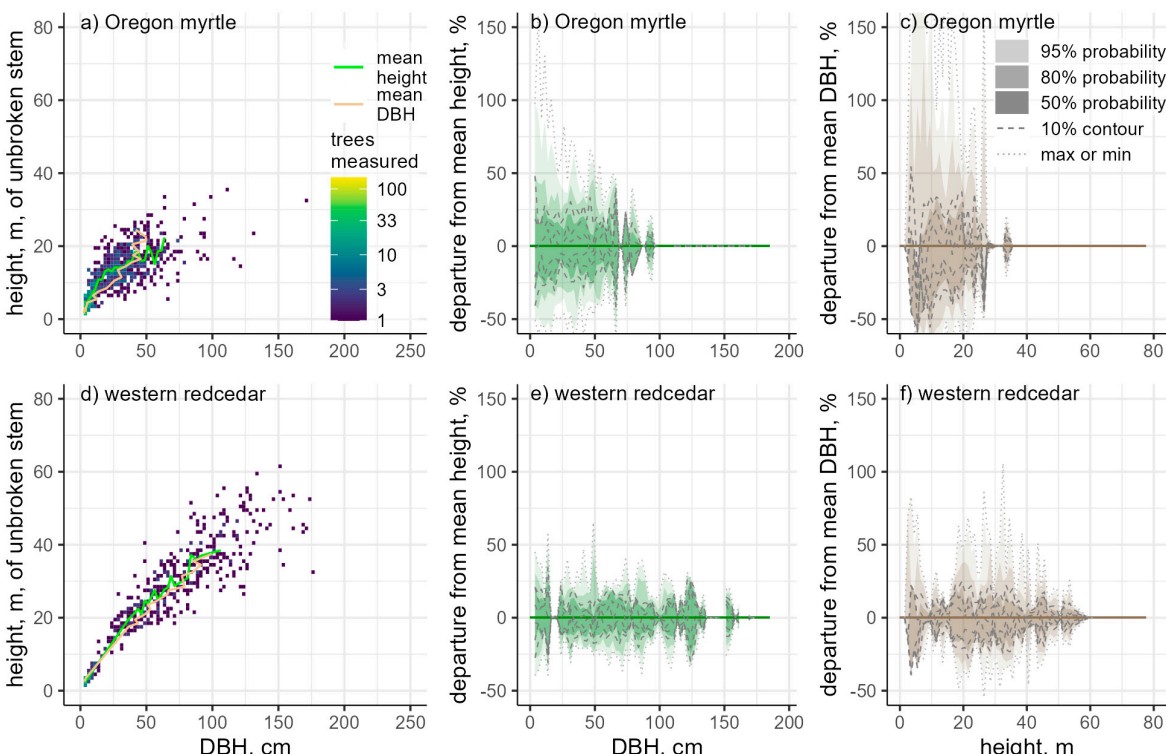

**Figure A3.** Distribution of unbroken (**a**–**c**) Oregon myrtle and (**d**–**f**) western redcedar stem sizes in 2015–2016 Elliott State Research Forest cruise data with respect to the trees' mean height (**b**,**e**) and diameter (**c**,**f**). Mean height and diameter lines are drawn as indicated in Figure A1.

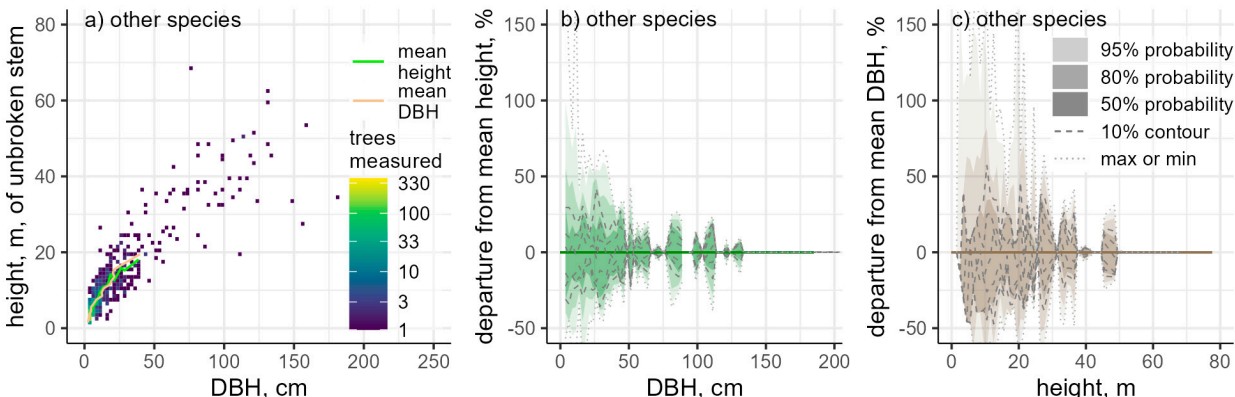

**Figure A4.** (**a**–**c**) Joint stem size distribution of all other tree species present in 2015–2016 Elliott State Research Forest cruise data (Table 1) that are not shown in Figures A1–A3. Mean height and diameter lines are drawn as indicated in Figure A1.

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
