# Peer review of "Elliott State Research Forest Timber Cruise, Oregon, 2015–2016"

_data, 2015_

Round 1
Reviewer 1 Report
Comments and Suggestions for Authors
Hello.
The paper presents a very valuable data set.
The presentation is very rich (even too rich) and interesting.
Presentation quality is good but there is large space for improvements.
In several points, phrase construction and terms used do not help to understand.
Some details concerning inventory design and cruising should be clarified.
Detailed comments are annotations in the attached pdf.
Regards

Comments on the Quality of English Language(in the attachment)
Reviewer 2 Report
Comments and Suggestions for Authors
I commend both of you greatly for the decision to make this kind of data publicly available, and I hope others in the forest inventory/biometrics field will begin to follow suit with your example. Data secrecy is one of the the factors that I believe has been holding back the field of forest biometrics from having higher relevance in the larger scientific community for a very long time.
I understand that holding back the plot location information is necessary at the present time because the culture of data secrecy is hard to change all at once. Perhaps you could comment in your paper about the reasons why this condition was put into place, and some of the suggestions you might make for how this final barrier to full openness can be removed.
One thing that you should very strongly consider, since it might be of interest to some of your data users, is to append interpolated climate information to the data. For example, the standard outputs from something like ClimateWNA:
https://sites.ualberta.ca/~ahamann/data/climatewna.html
Future users of your data may want to include climate co-variates into their modelling, and given that location can't be made available, it would be impossible for them to do it themselves at a later date.
